# The Use of Intravenous Lidocaine in Perioperative Medicine: Anaesthetic, Analgesic and Immune-Modulatory Aspects

**DOI:** 10.3390/jcm11123543

**Published:** 2022-06-20

**Authors:** Ingrid Wing-Sum Lee, Stefan Schraag

**Affiliations:** 1School of Medicine, University of Glasgow, Wolfson Medical School Building, University Avenue, Glasgow G12 8QQ, UK; 2362652l@student.gla.ac.uk; 2Department of Perioperative Medicine, Golden Jubilee National Hospital, Agamemnon Street, Clydebank G81 4DY, UK

**Keywords:** lidocaine, intravenous, analgesia, multi-modal anaesthesia, opioid-sparing

## Abstract

This narrative review provides an update on the applied pharmacology of lidocaine, its clinical scope in anaesthesia, novel concepts of analgesic and immune-modulatory effects as well as the current controversy around its use in perioperative opioid-sparing multi-modal strategies. Potential benefits of intravenous lidocaine in the context of cancer, inflammation and chronic pain are discussed against concerns of safety, toxicity and medico-legal constraints.

## 1. Introduction

Lidocaine is one of the most commonly used drugs in anaesthetics. It was first synthesised in 1942 under the name Xylocaine^®^ and subsequently approved for use in 1948 in Sweden [1]. Since 1977, it has been listed in the World Health Organisation’s Model List of Essential Medicines as a local anaesthetic and also for the management of ventricular tachyarrhythmia [2]. These essential medicines are known to have public health relevance in terms of efficacy, safety and cost-effectiveness [3]. As a local anaesthetic, lidocaine can be used for surface anaesthesia, infiltration anaesthesia, intravenous regional analgesia and nerve block and dental anaesthesia [4]. Historically, lidocaine has been used intravenously as an anti-arrhythmic agent [5]. First reports of lidocaine’s analgesic properties date back to the 1950s and 60s [6,7,8]. However, research has recently come to light suggesting intravenous lidocaine may have potential benefits in the perioperative setting. It may have a role in reducing both pain and postoperative nausea and vomiting (PONV), which are two of the most common complaints after surgery and anaesthesia. Therefore, it may prove to be useful in the improvement of postoperative pain and recovery outcomes. This review aims to explore the mechanism of action of intravenous lidocaine, its potential use in the perioperative setting, alongside its safety profiles and efficacy for different types of surgeries. In addition to its anaesthetic properties, lidocaine may also have immune modulatory properties which may be beneficial for cancer treatment, which this review also aims to briefly delve into.

## 2. Pharmacology and Mechanism of Action

Lidocaine is an amide type local anaesthetic with the molecular structure of C_14_H_22_N_2_O [5]. When administered intravenously it is 60–80% protein bound, mainly to the acute phase reactant alpha-1-glycoprotein [9]. Lidocaine crosses both the blood brain barrier and placenta through simple passive diffusion. It is also excreted in breast milk; hence, in order to avoid toxicity to the breast-fed infant, this needs to be taken into consideration in breastfeeding mothers [5]. 

The main metabolism of lidocaine is by oxidative N-dealkylation. This occurs mainly in the liver by the cytochrome p450 system, specifically the enzyme CYP3A4 [10]. Lidocaine is deethylated to its active metabolite monoethylglicinexylidide (MEGX), then to glycinexylidide (GX), and various other metabolites. The latter is further hydrolysed to other byproducts found in urine. Figure 1 below shows the chemical structure of lidocaine and its metabolites. 

The MEGX concentration has been used to evaluate liver function after lidocaine administration [11]. In terms of its anti-convulsant and anti-arrhythmic properties, MEGX has around 80% potency compared to that of lidocaine [12]; it also decreases the clearance of lidocaine [13]. Conversely, GX has much less activity and is metabolised and excreted by the kidney. MEGX and GX can cause toxicity in patients with cardiac failure and renal failure, respectively [13]. 

Lidocaine alongside its metabolites is renally excreted, with just under 10% excreted unchanged in urine [14]. The elimination half-life of lidocaine generally ranges from 90 to 120 min in healthy patients, and it can be prolonged in patients with coexisting illness such as hepatic or renal impairment or congestive cardiac failure. Additionally, its excretion is influenced by urinary pH [15,16].

A study by De Martin et al. [17] showed that lidocaine clearance is affected linearly with renal impairment, with a double elimination half-life when compared to patients with normal renal function. Likewise, in older patients, the elimination half-life is significantly longer compared to their younger counterparts; thus, elderly patients should be given a lower continuous infusion rate [18]. Lidocaine is also protein bound; hence, hypoalbuminaemia may predispose to the risk of toxicity due to the increased amount of free drugs available. Hepatic blood flow is another limiting factor in its metabolism, despite lidocaine being predominantly renally excreted [19].

## 3. Intravenous Use

According to the Vaughan Williams classification, lidocaine is a class Ib anti-arrhythmic drug [19]. Drugs in this class block sodium channels during phase 0 of the action potential. It is used intravenously as an anti-arrhythmic agent for cardiopulmonary resuscitation if amiodarone is unavailable or contraindicated [4]. Lidocaine has a distribution half-life of five to eight minutes, beginning at the vascular compartments, then into the peripheral tissues [5]. It first passes into well-perfused areas such as the heart and lung, followed by less-perfused areas such as muscle and adipose tissue [5]. The brain and heart have been found to have the highest blood concentrations of lidocaine compared to other organs, which likely explains the CNS and CVS toxicity associated with local anaesthetics such as lidocaine [20,21].

The efficacy of lidocaine as an anti-arrhythmic drug depends on its plasma concentration. An initial bolus of lidocaine can transiently suppress arrhythmia. However, to sustain this effect, a continuous infusion is required in order to achieve the therapeutic plasma concentration. Unfortunately, the longer the duration of infusion, the higher the likelihood of toxicity, as enzymes become saturated and clearance rates are decreased. This is seen in particular after 24 h, as confirmed by clinical trials [22]. Research suggests avoiding infusions of longer than 24 h duration and to give infusions based on the patient’s body weight.

As mentioned previously, lidocaine’s main mechanism of action is through the blockage of voltage gated sodium channels. By reducing the permeability of cell membranes to sodium, this in turn decreases membrane depolarisation, blocking the propagation of the action potential, and hence decreases the neural conduction of pain stimuli. At therapeutic concentrations during intravenous infusion, lidocaine blocks muscarinic M1 and M3 receptors, as well as NMDA receptors. At higher concentrations, lidocaine exerts its effects on a number of other receptors such as 5-hydroxytryptamine-3 (5HT-3), nicotinic cholinergic receptors, voltage-gated calcium channels (VGCC) and many others [23].

## 4. Clinical Properties of Lidocaine

This section describes various clinical domains which define the spectrum in which lidocaine is being considered as therapeutically beneficial.

### 4.1. Analgesic, Anti-Hyperalgesic and Anti-Nociceptive

Firstly, lidocaine exhibits analgesic, anti-hyperalgesic and anti-nociceptive properties. Eipe et al. [13] stated that lidocaine reduces the sensitivity and activity of spinal cord neurones, and also decreases NMDA receptor mediated postsynaptic depolarisation. 

There is no definitive single molecular mechanism of intravenous lidocaine as an analgesic; its perioperative analgesic effects are likely multi-factorial. It involves the inhibition of sodium, potassium and calcium channels, Gαq-coupled protein receptors and NMDA receptors, to name a few. The 5HT3 receptor may also be involved, as systemic ondansetron has shown to antagonise the sensory block produced by intrathecal lidocaine [24].

Kawamata et al. demonstrated that the injection of lidocaine prior to surgical incision reduces primary hyperalgesia more effectively when compared to injection after the incision [25]. Primary hyperalgesia is limited to the site of incision, with hyperalgesia to mechanical and thermal stimuli; secondary hyperalgesia, on the contrary, is noticed in intact skin surrounding the incision site, with hyperalgesia due to mechanical stimuli. Holthusen et al. [26] found that lidocaine did not affect primary hyperalgesia. However, in another study, researchers found that intravenous lidocaine can temporarily suppress primary hyperalgesia by stabilising peripheral nerves. Nevertheless, the effects of both primary and secondary hyperalgesia are thought to be due to peripheral and central sensitisation [27].

Remifentanil is a common opioid analgesic used intraoperatively to treat nociceptive pain. However, at high doses, it is associated with postoperative opioid-induced hyperalgesia [28]. A study by Cui et al. demonstrated in rats that the use of systemic lidocaine can reduce remifentanil-induced hyperalgesia through inhibiting conventional protein kinase C gamma (cPKCγ) membrane translocation [29]. Thus, intravenous lidocaine may be a good option to counteract this ‘opioid-induced’ hyperalgesia through its proposed anti-hyperalgesic effects. 

Numerous ion channels may be implicated in lidocaine’s analgesic properties. Voltage gated sodium channels (VGSCs) play an important role in nociceptive signalling and sensory transmission. Research has discovered nine different isotopes of VGSCs, of which six are expressed in the dorsal root ganglion, which is involved in neuropathic and inflammatory pain pathways [30]. Potassium channels come in multiple types, of which voltage gated (Kv) subunits and tandem pore domain (K2P) channels are involved in pain modulation [9]. Further, lidocaine increases intracellular calcium concentration. Modulation of calcium is involved in mechanisms underlying neuropathic pain [31,32]. Low voltage-activated T type calcium channels, specifically the Ca_V_3.2 subtype, are involved in both somatic and visceral pain [33].

### 4.2. Anti-Arrhythmic

Lidocaine, administered intravenously, is mainly used as an anti-arrhythmic agent. Lidocaine 100 mg IV is included as part of the latest European Resuscitation Council Adult Advanced Life Support guidelines as an alternative to amiodarone for patients in ventricular fibrillation or pulseless ventricular tachycardia; an additional 50 mg bolus may also be given after five defibrillation attempts [34]. It decreases the slope of phase 4 in the action potential and changes the excitability threshold, resulting in a decrease in action potential length and duration of refractory period of Purkinje fibres [19]. Despite this, the caveat to consider is that lidocaine itself can predispose patients to arrhythmias, especially at high doses or concentrations. On the contrary, a recent case report in 2020 suggested that intravenous lidocaine dampened QT prolongation when given with azithromycin and chloroquine/hydroxychloroquine [35].

### 4.3. Anti-Inflammatory

Lidocaine is known to have anti-inflammatory effects; although, the exact mechanism remains unclear. Lidocaine can inhibit leucocyte activation, adhesion and migration [36]. It also protects cells from inflammation through the reduction of neutrophil adhesion and inhibition of the release of superoxide anions [36,37]. It has also been documented to block the release of inflammatory mediator interleukin-1B in in vitro studies [38]. Lidocaine’s direct effects on macrophage and polymorphonuclear granulocyte functions may also contribute to its anti-inflammatory effects in addition to the inhibition of release of interleukins involved in the inflammatory cascade. Further, it also inhibits prostaglandin biosynthesis and release, which is another possible explanation for its powerful anti-nociceptive and anti-inflammatory actions [19].

In vivo studies have also shown lidocaine at high concentrations to inhibit histamine release from human leucocytes, mast cells and basophils [39]. Studies on murine models showed that the level of pro-inflammatory markers remained low in the group treated with intravenous lidocaine [40]. Unfortunately, human studies demonstrating the anti-inflammatory properties of lidocaine are limited. A number of studies investigating abdominal and colorectal surgeries such as laparoscopic cholecystectomy, have shown that lidocaine’s use in the perioperative setting has reduced the surgery-induced release of pro-inflammatory cytokines, for example, IL-6 and IL-8 [41,42,43].

A recent study in 2022 [44] investigated the effect of local anaesthetics on tumour necrosis factor-alpha (TNF-α) secretion. TNF-α normally plays an important role in inflammation and carcinogenesis. The study found that with the use of lidocaine, 61.5% of individuals demonstrated an ≥85% reduction of TNF-α-production of lipopolysaccharide-activated human leucocytes. Hence, lidocaine may possibly be an option for treating chronic inflammation or conditions with overactive immune responses, for example, acute respiratory distress syndrome (ARDS), though further studies are needed to explore this further.

## 5. Evidence on Postoperative Outcome

In terms of improving postoperative recovery outcomes, a number of studies have investigated the effect of intravenous lidocaine on different surgeries. Relatively speaking, abdominal surgery has been studied most in comparison to other surgeries, and will be discussed briefly below. 

### 5.1. Colorectal Surgery

Around 40% of patients experience a delay in resumption of normal bowel function after colorectal surgery. This delay leads to symptoms of nausea, vomiting, constipation and abdominal distension, which then require unpleasant supportive interventions such as intravenous fluids and nasogastric tube insertion. There is no remedy to address this delay. ALLEGRO, “A placebo-controlled rAndomised trial of intravenous Lidocaine in acceLErating Gastrointestinal Recovery after cOlorectal surgery,” is the latest ongoing multi-centre research study across the United Kingdom, investigating the use of intravenous lidocaine to improve recovery after colorectal surgery [45]. Evidence corroborated from various meta-analyses in the past have shown that perioperative lidocaine infusion at doses of 1.5 to 3 mg/kg/h consistently improved postoperative Visual Analogue Scale (VAS) pain scores in patients undergoing either open or laparoscopic abdominal surgery [46,47,48]. However, it is important to note that the recommended dose of continuous lidocaine infusion should be no more than 1.5 mg/kg/h [23]. Intraoperative and postoperative opioid requirements were also decreased with the use of lidocaine as an inpatient [48]. Further, perioperative intravenous lidocaine may benefit bariatric patients more as they are more sensitive to respiratory depression caused by opioids [49]. In addition to reducing pain, another benefit is a reduction in the duration of postoperative ileus by an average of eight hours [47,50]. Further, decreased postoperative nausea and vomiting of up to 20% were reported, likely because opioid consumption was decreased [47,51]. Intravenous lidocaine also reduced the length of hospital stay by an average of eight hours and up to 24 h at most [46,47,51]. Lidocaine’s use has been reported to have slightly lower pain scores [52] at one to four hours after surgery, alongside a reduction in postoperative ileus duration and time to return of gastrointestinal function-time to first flatus and first bowel movement. From the limited evidence above, it may be easy to presume perioperative lidocaine is a useful analgesic adjunct to general anaesthesia in colorectal surgery. However, when lidocaine is compared to another intervention such as epidural analgesia, one study reported that intravenous lidocaine was inferior to epidural analgesia in major abdominal surgery with regards to total opioid consumption; although, it did improve other aspects of recovery [53]. This demonstrates limitations in the evidence, with the main one being publication bias, where positive results in favour of lidocaine are more likely to be published. Further, in many papers, namely the Cochrane Systematic Review [52], the trials reviewed mainly used placebo as the control, which naturally makes a clinical effect more likely to be detected, as opposed to studies which compare lidocaine to an alternate therapy, such as one compared to epidural analgesia. Whilst the review gathers numerous trials, the evidence remains of low quality due to the heterogeneity of the results. In addition, the mechanism of action of lidocaine is extremely complex and there is no single established molecular mechanism. A long list of receptors may be implicated, and these individual receptors themselves have different levels of sensitivity which are further influenced by many factors such as inflammation, trauma and concomitant medications. Thus, all the evidence needs to be considered with the utmost care.

### 5.2. Other Surgical Specialties

Perioperative lidocaine was also shown to improve pain and reduce opioid consumption for patients undergoing radical retropubic prostatectomy [54]. However, there was little benefit for patients undergoing laparoscopic renal surgery [55]. Likewise, with cardiothoracic surgery, lidocaine has little benefit in patients undergoing cardiac surgery [56] but may have value for patients undergoing thoracic surgery. One systematic review [57] investigated lidocaine’s effects on postoperative pain and recovery after cardiac surgery. Insler et al. [56] carried out a randomised double-blinded trial of continuous low-dose lidocaine intravenous infusions in coronary artery bypass graft (CABG) patients. Patients in the trial were given an 8 mg/mL lidocaine infusion after induction with fentanyl and a mixture of fentanyl and midazolam for maintenance. The study concluded that the lidocaine infusion did not decrease the length of ICU and hospital stay or time to extubation, and also did not significantly decrease the use of other opioid analgesics [56]. On the other hand, the lidocaine infusion reduced pain scores up to six hours after thoracic surgery [58].

Evidence for obstetric and gynaecologic surgery is limited and outcomes are mixed [59]. Further, a randomised double-blind placebo-controlled study of perioperative lidocaine infusion for patients undergoing bariatric surgery found no difference clinically in terms of postoperative outcomes including pain, nausea and vomiting, length of stay and oxycodone consumption [60]. On a different note, for breast surgery, lidocaine’s short-term benefit is limited; however, it may prove to be more useful long term as it has been reported to reduce the incidence of chronic postsurgical pain at three and six months after mastectomy [61].

## 6. Immuno-Modulatory and Anti-Cancer Properties

Recent research has explored the possibility of perioperative lidocaine in improving cancer outcomes, in particular its role in reducing the recurrence of metastatic cancer [62]. Cancer is a major public health and economic burden worldwide. Many patients undergo surgery as part of their cancer diagnosis or treatment. An unavoidable consequence of surgery is that cancer cells are dislodged, which provokes a physiological stress response comprised of inflammation and angiogenesis. The surgical stress response is essential to promote wound healing. Sadly, cancer is ‘a wound that does not heal’ [63]. Growing evidence suggests that the surgical stress response may paradoxically facilitate the survival and replication of residual cancer cells postoperatively, potentially dislodging cancer cells; the formation of circulating tumour cells later metastasise to distant organs [64]. Laboratory research has demonstrated that common anaesthetic drugs, for example, lidocaine, may have anti-neoplastic effects. Reports dating as far back as 1982 [65] first suggested procaine and lidocaine combined with doxorubicin enhanced doxorubicin cytotoxicity against a human melanoma cell line derived from malignant ascites. Since then, much more laboratory evidence has come to light, suggesting lidocaine may exert its potential anti-neoplastic effects via multiple pathways, in addition to other anti-inflammatory effects. 

A lot of research has gone into studying the effects of lidocaine on breast cancer. D’Agostino et al. [66] showed that clinical concentrations of lidocaine inhibit CXCL12-induced CXCR4 signalling, which inhibits calcium release and actin polymerisation. This in turn impairs cytoskeleton remodelling, and hence reduces the migration of breast cancer cells. Another study suggested that lidocaine only affected cell viability or migration at high or toxic concentrations via the arrest of cancer cells in the S phase [67].

Li et al. suggested that lidocaine can act as a chemosensitiser for cancer treatment [68]. Lidocaine enhanced apoptosis and sensitised cisplatin to a highly aggressive triple negative breast cancer. Further, treatment with lidocaine caused the suppression of Ras Association Domain Family 1A (RASSF1A) and Retinoic acid receptor β (RARβ2) gene methylation, which is the mechanism by which lidocaine sensitises cisplatin to breast cancer cells [68]. Another chemotherapeutic agent sensitised by lidocaine is 5-fluorouracil (5-FU). Its combination with 5-FU sensitised melanoma cells to 5-FU via upregulation of “miR-493 and the down-regulation of Sox4- mediated PI3K/AKT and Smad pathways” [69]. Another paper reported that lidocaine acts as a chemo-sensitiser to 5-FU through decreasing viability, increasing apoptosis and downregulating expression of ATP-binding cassette (ABC) transport proteins [70].

A study in 2019 indicated lidocaine’s role in suppressing the metastasis of breast cancer via the suppression of pro-inflammation factors [71]. The study compared the effect of perioperative lidocaine, propofol and steroids in breast cancer surgery. Results of the study showed that both lidocaine and propofol could reduce pulmonary metastasis from breast cancer, whereas steroids actually increased metastasis. Another study found that intravenous perioperative lidocaine for breast cancer surgery decreased the postoperative expression of neutrophil extracellular trapping (NETosis), which is a mechanism linked to increased metastatic risk. This may support the hypothesis that the use of intravenous lidocaine for cancer surgery may reduce recurrence [72]. 

Other mechanisms of metastasis inhibition may be attributed to lidocaine’s anti-inflammatory and anti-angiogenic effects, according to a study in 4T1 breast cancer cell line in vitro and in vivo [73]. Taking all the evidence into account, all these studies point to lidocaine’s potential role as an “anti-cancer drug”, so to speak. Fraser et al. also demonstrated that lidocaine was shown to inhibit VGSC in breast cancer cells [74]. VGSCs are expressed in active breast, colon and prostate cancers [74]. By inhibiting VGSCs, this reduces cellular activity, hence leading to reduced cell division. 

In lung cancer, in vitro studies demonstrated that lidocaine reduces ICAM-1 and Src phosphorylation after the stimulation of tumour necrosis factor (TNF) [75]. By reducing ICAM-1 activity, this may inhibit tumour cell adhesion to vascular endothelium, hence preventing migration. It also suppressed matrix metalloproteinase 9 (MMP-9) secretion significantly; another pointer to its potential role in inhibition of cancer cell invasion and metastasis [75]. In addition to its anti-tumour effects, postoperative atrial fibrillation is a common finding post-lung cancer surgery, and lidocaine has shown activity in suppressing this [76].

In human hepatocellular carcinoma cells, lidocaine arrested the growth of HepG2 cells and also induced apoptosis, possibly through an increase in Bax protein and activated caspase-3 and decreasing Bcl-2 protein via extracellular signal-regulated kinase ½ and p38 pathways; it also enhanced the sensitivity of cisplatin [77]. On the other hand, another study showed that lidocaine at concentrations of 1.25 to 5 mg/mL inhibited proliferation of bladder cancer cells in a concentration-dependent manner. When lidocaine was combined with other chemotherapy agents such as mitomycin C, it also enhanced actions of these antiproliferative agents [78]. Both of the aforementioned studies included both in vitro and in vivo animal studies.

To summarise, numerous mechanisms have been proposed in the research of how local anaesthetics such as lidocaine are thought to reduce cancer recurrence [79]. Such mechanisms include but are not limited to (Figure 2):

The effects of local anaesthetics on tumour progression can be further classified into indirect and direct effects, as shown below in Figure 3.

Figure 4 below is a compilation of the vast array of mechanisms by which lidocaine exerts its effects on cancer cells.

Despite the many studies, limitations are present which preclude definitive conclusions to be drawn regarding the effect of lidocaine in reducing cancer metastasis and recurrence. The plethora of evidence and studies demonstrate a multitude of potential mechanisms by which lidocaine may act as an immune-modulating drug. However, each cancer has different cell types which are unique in their function; thus, no clear consensus can be made regarding the exact mechanism by which lidocaine may demonstrate a therapeutic effect. Further research is needed in order to fully translate laboratory findings into clinical implications.

## 7. Clinical Recommendations and Safety

With regards to lidocaine being used intravenously in the perioperative setting, there are multiple reasons it may be used. One such reason is its use in reducing pain from the propofol injection. Propofol-induced injection pain is a common occurrence during the induction of anaesthesia with propofol. Whilst the underlying mechanism for this pain is unclear, it is likely due to a combination of nociceptor stimuli and release of pain mediators such as bradykinin. Studies have shown that lidocaine given prior to propofol injection, either mixed with propofol or given separately, with or without venous occlusion, reduced post-injection pain. Its exact mechanism in reducing pain remains unclear [80].

An international consensus statement was published in 2021 regarding the use of intravenous lidocaine [23]. Current recommendations, broadly speaking, err on the side of caution, as intravenous lidocaine remains unlicensed for analgesia. Like most medical decisions, explicit informed consent from patients should be obtained where possible before its use, so as to minimise any potential for legal disputes should they arise in the future. Doctors need to take into account any absolute or relative contraindications, such as renal or hepatic impairment, cardiac disease or conduction block, seizure disorders, electrolyte imbalances, pregnancy, breast feeding or neurological disorders [81,82]. The British National Formulary (BNF) also lists the following contraindications for intravenous use: atrioventricular blocks, severe myocardial depression and sino-atrial disorders. Other cautions to be aware of include acute porphyria, congestive cardiac failure and postcardiac surgery [4]. 

The exact dose of intravenous lidocaine to be used is debatable. The calculation of the maximum recommended dose is based on the patient’s ideal body weight; although, this is just a guide [83]. One study demonstrated that dosing based on actual body weight resulted in 20% higher than predicted plasma concentrations, thus resulting in an increased potential for toxicity [84]. Systemic lidocaine should be avoided altogether in patients weighing under 40 kg and the maximum dose for any patient should not exceed 120 mg/h. The initial loading dose should be a maximum of 1.5 mg/kg initially over 10 min, followed by a continuous infusion of 1.5 mg/kg/h, with continuous reassessment with ECG and blood pressure monitoring and pulse oximetry [23], according to the consensus statement. Conversely, others have suggested that when IV lidocaine is started in theatre or critical care areas, therapeutic levels of 2.5–3.5 µg/mL may be maintained on regular wards with no need for continuous ECG monitoring [13]. The consensus statement [23] adopts a safer approach, which recommends patients receiving IV lidocaine to be managed in a monitored bedspace such as a high dependency unit (level 2 care). Another guideline from Imperial College Healthcare NHS Trust states that “ECG monitoring should be continuous while any patient remains on an IV lidocaine infusion [85].” Postoperatively, the minimum frequency of observations such as sedation score, BP, HR, RR and SpO2 should be carried out every 15 min in the first hour, then every half hourly for two hours, and hourly thereafter. 

Lidocaine should not be infused for over 24 h due to the risk of toxicity. Most patients will have recovered sufficiently after 24 h as systemic lidocaine is not the only analgesic being used. As a safety measure, lipid emulsion 20% should be readily available for emergencies. In patients with an increased risk for toxicity, plasma lidocaine levels may be monitored as an extra safety precaution [23].

The most commonly reported clinically effective dose of lidocaine infusion ranges from 1 to 2 mg/kg/h. This is in keeping with the consensus statement mentioned previously. Continuous infusion requires four to eight hours to achieve steady state plasma concentration; therefore, eight hours should be allowed to achieve steady state before dose adjustments are made [85]. Generally, there is no accumulation of lidocaine in healthy individuals.

Lidocaine has a narrow therapeutic index, with a therapeutic plasma level of 2.5 to 3.5 μg/mL [13]. Central nervous system (CNS) toxicity occurs when plasma concentration exceeds 5 μg/mL and is definite at 10 μg/mL [13]. Patients who are awake typically have predictable symptoms. Early symptoms include perioral paraesthesia and metallic taste [14], followed by light-headedness and tinnitus. This progresses to muscle twitching, a reduced level of consciousness and seizures. Toxicity, if left untreated, may lead to respiratory depression and apnoea, or even cardiovascular collapse and coma [13]. The higher the lipid solubility of a drug, the higher the risk of cardiotoxicity. Signs of cardiac toxicity include but are not limited to arrhythmias, hypertension, hypotension, bradycardia and conduction block.

Lidocaine toxicity more commonly manifests as neurological symptoms rather than cardiovascular signs. In general, the cardiovascular system appears to be more resistant to local anaesthetic effects compared to the CNS. However, cardiovascular toxicity remains a significant adverse effect of local anaesthetic systemic toxicity. Signs of cardiac toxicity occur when serum levels exceed 10 μg/mL [13]. All signs and symptoms of toxicity are potentiated by acidosis, hypercapnia and hypoxia. Acidosis increases the risk of toxicity because lidocaine dissociates from plasma proteins; hence, hypoalbuminaemia may also predispose to toxicity [19].

The exact doses or concentrations at which toxicity manifests are difficult to predict in patients; thus, careful continuous monitoring is required. Intravenous lidocaine is also contraindicated if other local anaesthetic interventions are used concurrently, for example, in various neuraxial blocks. The exact timing of when it can be used remains unclear. Foo et al. [23] recommended lidocaine not to be used specifically within four hours of interventions. However, some authors believe this four-hour rule to be problematic [86]. It is easy to assume a straightforward additive effect when multiple local anaesthetics are combined. However, it is likely more complex, because each patient has a slightly different physiology, and different anatomical structures may have different anaesthetic requirements, alongside different pharmacokinetic variables. As clinicians, there is the tendency to assume multiple interventions automatically translate to additional benefits; however, this may in fact increase the risk of harm to patients. Therefore, multiple interventions of local anaesthetics (MILANA) should be avoided in susceptible populations [86]. Some strategies to minimise local anaesthetic systemic toxicity (LAST) are shown in Figure 5:

Despite the lack of guidelines, intravenous lidocaine is widely used. A survey in 2020 of Australian and New Zealand anaesthetists found that more than 50% of respondents used lidocaine intravenously [87]. In addition, a survey of 16 Scottish hospitals [88] found that 12 hospitals (75%) either use or plan to use intravenous lidocaine for the management of acute pain. These hospitals have some sort of established or provisional guideline. One hospital stated that intravenous lidocaine is briefly mentioned in the Enhanced Recovery After Surgery (ERAS) patient information leaflet, but there is no further information. In total, 50% of hospitals only allowed consultants to prescribe lidocaine infusions; a small number allowed specialty trainees or acute pain nurses to prescribe; and one hospital allowed any member of the medical staff to prescribe the infusions. Across the Scottish hospitals surveyed, there was considerable variability in the maximum duration of intravenous lidocaine infusion, ranging from 4 h to 72 h. Regarding the concurrent use of local anaesthetic infusion, 1 hospital used the four-hour waiting rule, whereas 5 out of 12 hospitals had no specific rule. In the United States of America, for example, the University of Virginia [89], IV lidocaine is routinely used for analgesia for acute pain management and also intraoperatively for various types of surgery; the decision for postoperative use is made on a case-by case-basis. They also do not require additional monitoring other than the standard protocol. The huge variability in protocols and guidelines across different countries emphasises the need for national guidelines to ensure safety if intravenous lidocaine becomes a part of everyday practise. 

## 8. Medico-Legal Implications and Licensing

In response to the consensus statement published by Foo et al. [23], an editorial [90] was published which highlighted the dangers of using intravenous lidocaine and the limited evidence regarding its clinical efficacy. In the United Kingdom, the Safe Anaesthesia Liaison Group (SALG), and its parent organisation the Royal College of Anaesthetists and its Faculty of Pain Medicine, do not endorse the recommendations in the consensus statement [90]. The literature is sparse with regards to toxicity from intravenous lidocaine, whereas reports of toxicity from local or topical injection is much more readily available in comparison. Supporters of the use of intravenous lidocaine argue that paradoxically intravenous use is safer, as practitioners are more alert to potential side effects. However, issues of using intravenous lidocaine as an unlicensed medicine remain. For example, if a continuous intravenous lidocaine infusion is used, there is concern that practitioners who did not commence the drug may be held responsible if any untoward event occurred. In addition, as there is no specific infusion pump designed for intravenous lidocaine, using existing medical devices to administer lidocaine would be considered “off-label”, and hence the Hospital Trust would assume full legal liability for any malfunction or misuse [90]. 

From the clinical evidence, the Cochrane Review of 68 trials with 4525 patients [52] ruled out any beneficial effect of lidocaine infusion at 24 h and 48 h post-surgery, despite a bias in favour of the “drug of interest”, that is, lidocaine. Only at 1 to 4 h post-operatively might there be some benefit, but this is still uncertain. Similarly, Weibel et al. [91] also recognised that the evidence for intravenous lidocaine in improving abdominal surgery outcomes is limited, and the best dose to be used remains uncertain. The clinical evidence, legal implications alongside the fine therapeutic-toxic index of lidocaine, causes one to seriously consider whether the use of this drug is appropriate. The Medical Devices, Medicines and Healthcare Products Regulatory Agency (MHRA) does not support the use of intravenous lidocaine for analgesia; manufacturers do not approve of its off-label use [90].

## 9. Summary and Conclusions

Having gathered a multitude of evidence on this contentious topic, the verdict remains unclear. Studies have shown that the use of intravenous lidocaine in the perioperative setting may be beneficial depending on the surgical procedure. Current limited evidence demonstrates a very small benefit for patients undergoing abdominal surgeries. “Primum non nocere” remains the central tenet of medicine. Prior to any medical decisions, it is essential for the risks and benefits to be weighed by the physician. Ultimately, it is at the individual physician’s discretion whether or not to use intravenous lidocaine. The physician must take responsibility for any possible harm which may befall a patient. To mitigate potential harm to the patient, screening for comorbidities, careful monitoring, correct dosing and having emergency drugs available: these are all crucial steps to be undertaken if intravenous lidocaine is used. 

In conclusion, intravenous lidocaine has the potential to play a pivotal role as a non-opioid analgesic adjunct in perioperative medicine. As the concept of multi-modal analgesia continues to evolve in the future, lidocaine is a cost-effective and readily available option to minimise the excessive use of opioids in the management of postoperative pain. However, at present, it is important to recognise that lidocaine itself is not a panacea. Current evidence for both its clinical efficacy and safety profile still remains weak. Given the risks, it is unclear if there is a clinically significant benefit for its use. Most important of all, our decisions should be patient-specific and procedure-specific. In addition to lidocaine’s analgesic role, there is limited evidence for using lidocaine as an “anti-cancer” drug, but there is room to explore this further in the future. At present, it is unlikely lidocaine will be used as a stand-alone “anti-cancer” drug, but it may serve as “chemotherapy synergists,” so to speak. More studies are needed to substantiate the use of intravenous lidocaine in clinical settings, especially with regards to its anti-inflammatory and potential immune modulating properties. 

## Figures and Tables

**Figure 1 jcm-11-03543-f001:**
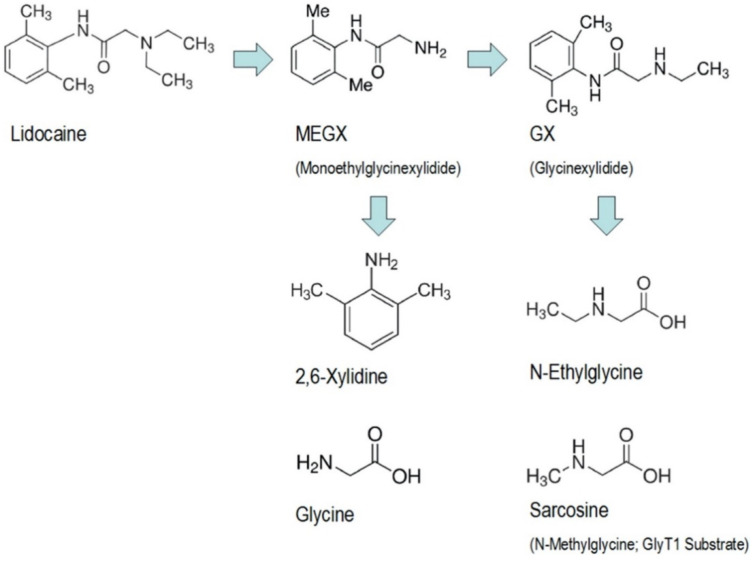
Chemical structure of lidocaine and its metabolites. Reprinted from Ref. [9], Copyright 2019, with permission from Elsevier.

**Figure 2 jcm-11-03543-f002:**
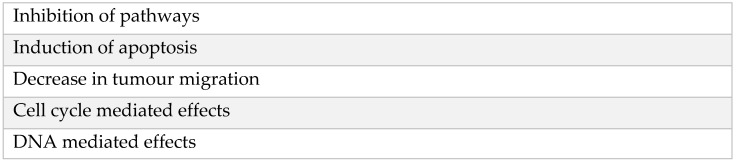
Proposed mechanisms of how lidocaine reduces cancer recurrence.

**Figure 3 jcm-11-03543-f003:**
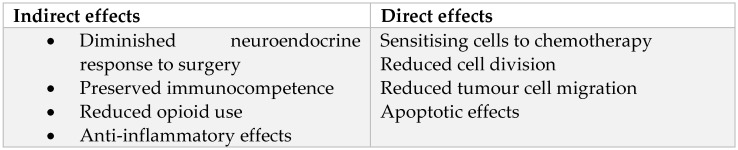
Indirect versus direct effects of local anaesthetics on tumour progression.

**Figure 4 jcm-11-03543-f004:**
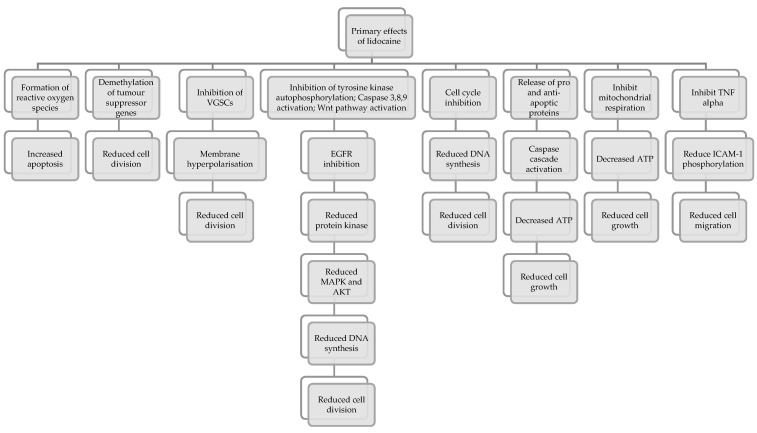
Summary of the actions of lidocaine on cancer cell activity, adapted from [79].

**Figure 5 jcm-11-03543-f005:**
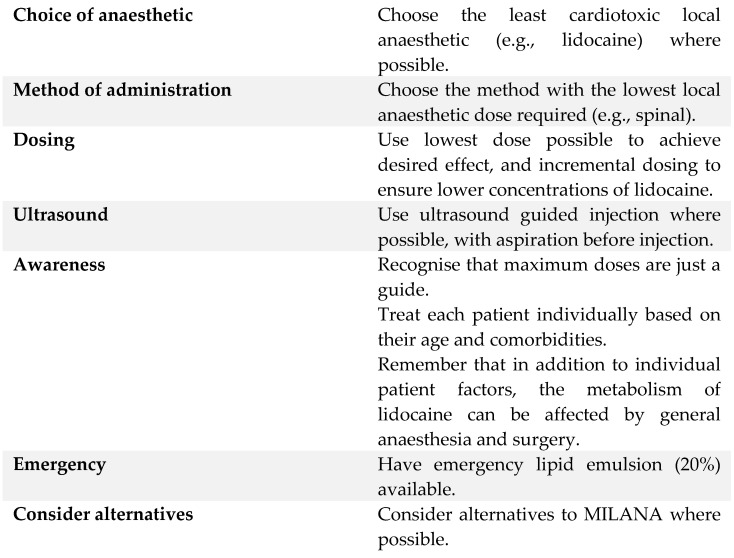
Strategies to minimise local anaesthetic systemic toxicity (LAST) [86].

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
