# Peer review of "The Use of Intravenous Lidocaine in Perioperative Medicine: Anaesthetic, Analgesic and Immune-Modulatory Aspects"

_jcm, 2022, doi:10.3390/jcm11123543_

Round 1

Reviewer 1 Report

Thank you for the opportunity to review this manuscript. This is an extensive review regarding the use of intravenous lidocaine in the perioperative setting. It focuses on the proposed mechanisms lidocaine may have on analgesia, cancer prevention, and PONV. My comments are below.

  1. The manuscript touches upon the opioid sparing effects secondary to lidocaine use. Could you please be more specific. Does this statement only apply to inpatient opioid sparing effects or does this carry over into the outpatient follow-up setting? Surgery is a risk factor for patients to develop new, persistent opioid use (ranging from 4-20% depending on the surgery), if it were to prevent continued outpatient opioid prescribing then there might be a bigger push to utilize lidocaine in the perioperative setting.
  2. The figures you have created provide a nice summary of what you have previously described, but are better suited for a PowerPoint presentation than a manuscript. Would suggest adjusting figures.
  3. Please review Figure 4 and be consistent with capitalization. 
  4. Authors had a nice section discussing lidocaine monitoring and levels. May want to include that many references do not suggest a need to monitor lidocaine levels when used at low doses for analgesia, especially if the lidocaine is only continued for 24 hours or less. Would also include that levels should be drawn at stead state before being interpreted unless signs of toxicity are present. 

Author Response

Changes are highlighted in yellow on the manuscript.

  1. The manuscript touches upon the opioid sparing effects secondary to lidocaine use. Could you please be more specific. Does this statement only apply to inpatient opioid sparing effects or does this carry over into the outpatient follow-up setting? Surgery is a risk factor for patients to develop new, persistent opioid use (ranging from 4-20% depending on the surgery), if it were to prevent continued outpatient opioid prescribing then there might be a bigger push to utilize lidocaine in the perioperative setting.

Authors response: Clarification that both intraoperative and postoperative opioid requirements are decreased as an inpatient. [please see Section 5.1] From what I have gathered I have not found studies specifically mentioning outpatient opioid requirements.

  1. The figures you have created provide a nice summary of what you have previously described, but are better suited for a PowerPoint presentation than a manuscript. Would suggest adjusting figures.

Authors response: Regarding style of figures: Amended as per comments. [please see new figures]

  1. Please review Figure 4 and be consistent with capitalization. 

Authors response: Figure 4: Amended capitalisation.

  1. Authors had a nice section discussing lidocaine monitoring and levels. May want to include that many references do not suggest a need to monitor lidocaine levels when used at low doses for analgesia, especially if the lidocaine is only continued for 24 hours or less. Would also include that levels should be drawn at stead state before being interpreted unless signs of toxicity are present. 

Authors response: Included reference which suggests continuous ECG monitoring not required. Also added reference about monitoring requirements during continuous IV lidocaine infusion. [please see Section 7]. Added point about allowing steady state to be achieved prior to dose adjustment. [please see Section 7].

Reviewer 2 Report

Thank you much for inviting me to review this very interesting manuscript. The topic is highly interesting and actual. There are many ongoing studies on this topic and many unanswered questions yet. I consider manuscript of interest because of extensive approach on lidocaine from mechanisms of action to safety and medical-legal aspects.

Manuscript is well written and argued  with references.

However I do have some minor suggestions.

1. Paragraph 3.2 Anti-arrythmic effects. A few details may be added: how is lidocaine administered and in what concentration for what kind of arrythmias. Guidelines on resuscitation may also be cited.

2. Paragraph 3.3. Along with reference 35 there are other references on this topic. Ex. Weinschenk 2022 Int J Mol Sci. 

3. Paragraph. 4.1 Comments on the 3mg/kg/h must e added because this is not the usual recommended dose for continuous infusion (max 1.5 mg/kg/h for longer terms infusions). This is discussed later in the manuscript but perhaps it would be of help to be discussed here as well. Comments on recommendations for laparoscopic surgery may be added since most of recommendations include this and epidural and no longer recommended for most such interventions. Literature on open abdominal surgery may also be provided and discussed in comparison with laparoscopic.

Chapter 5. Reference Galoș EV, Tat TF, Popa R, Efrimescu CI, Finnerty D, Buggy DJ, Ionescu DC, Mihu CM. Neutrophil extracellular trapping and angiogenesis biomarkers after intravenous or inhalation anaesthesia with or without intravenous lidocaine for breast cancer surgery: a prospective, randomised trial. Br J Anaesth. 2020 Nov;125(5):712-721. may be added since this is the first study on lidocaine and NETosis in breast cancer.

A short paragraph on lidocaine and hepatocarcinoma and bladder cancer may also be added since there are very interesting in vitro and animal studies on this types of cancers.

Fig 3. Direct effects of lidocaine also include apoptotic effects. 

Chapters 6 and 7. A short phrase on differences between countries/systems on regulations on lidocaine may be added.

Best regards

Author Response

Revisions are highlighted in green in the manuscript.

1. Paragraph 3.2 Anti-arrythmic effects. A few details may be added: how is lidocaine administered and in what concentration for what kind of arrythmias. Guidelines on resuscitation may also be cited.

Authors response: Regarding anti-arrhythmic properties: Added reference and info on IV lidocaine in resuscitation guidelines. [please see section 4.2].

2. Paragraph 3.3. Along with reference 35 there are other references on this topic. Ex. Weinschenk 2022 Int J Mol Sci.

Authors response: Regarding section on anti-inflammatory properties: [please see section 4.3].

3. Paragraph. 4.1 Comments on the 3mg/kg/h must e added because this is not the usual recommended dose for continuous infusion (max 1.5 mg/kg/h for longer terms infusions). This is discussed later in the manuscript but perhaps it would be of help to be discussed here as well. Comments on recommendations for laparoscopic surgery may be added since most of recommendations include this and epidural and no longer recommended for most such interventions. Literature on open abdominal surgery may also be provided and discussed in comparison with laparoscopic.

Authors response: Addressed maximum recommended dose of 1.5mg/kg/h for longer term infusions, as some studies included infusions of up to 3 mg/kg/h. [please see Section 5.1].

4. Chapter 5. Reference Galoș EV, Tat TF, Popa R, Efrimescu CI, Finnerty D, Buggy DJ, Ionescu DC, Mihu CM. Neutrophil extracellular trapping and angiogenesis biomarkers after intravenous or inhalation anaesthesia with or without intravenous lidocaine for breast cancer surgery: a prospective, randomised trial. Br J Anaesth. 2020 Nov;125(5):712-721. may be added since this is the first study on lidocaine and NETosis in breast cancer.

Authors response: Added reference Galos et al on lidocaine decreasing NETosis in breast cancer. [please see Section 6].

5. A short paragraph on lidocaine and hepatocarcinoma and bladder cancer may also be added since there are very interesting in vitro and animal studies on this types of cancers.

Authors response: Added paragraph on lidocaine and hepatocellular carcinoma and bladder cancer, with new references. [please see section 6].

6. Fig 3. Direct effects of lidocaine also include apoptotic effects.

Authors response: Figure 3 – added apoptotic effects to Figure on ‘direct effects of lidocaine’.

7. Chapters 6 and 7. A short phrase on differences between countries/systems on regulations on lidocaine may be added.

Authors response: Added brief section on America (having already spoken about Australia, New Zealand and Scotland). [please see end of section 7].

Short summary statement on variability of guidelines between different countries [please see end of section 7].